# Food Addiction Problems in College Students: The Relationship between Weight-Related Variables, Eating Habits, and Food Choices

**DOI:** 10.3390/ijerph192114588

**Published:** 2022-11-07

**Authors:** Sónia Gonçalves, Sílvia Félix, Filipa Martins, Olívia Lapenta, Bárbara C. Machado, Eva M. Conceição

**Affiliations:** 1Psychology Research Center (CIPsi), School of Psychology, University of Minho, 4710-057 Braga, Portugal; 2Research Centre for Human Development (CEDH), Faculty of Education and Psychology, Universidade Católica Portuguesa, 4150-268 Porto, Portugal

**Keywords:** food addiction, weight dissatisfaction, eating habits, food choices

## Abstract

The concept of food addiction, characterized by a strong urge to overeat highly palatable foods, has gained increased research attention over the last decade. College students are a recognized risk group for manifesting an eating pathology and weight gain due to the changes in eating habits experienced during this period. However, there is a gap in the literature connecting food addiction with eating and weight variables in this population. Thus, the present study aims to characterize food addiction in a sample of college students and enlighten the relationship between food addiction, weight-variables, eating habits, and food choices in this population. A sample of 194 college students (89.2% females) aged between 18 and 32 years old (M = 20.85, SD = 2.78) completed a set of self-reported online questionnaires on Google Forms. Namely, a Sociodemographic and Anthropometric Questionnaire, a questionnaire on Food Choices Characterization, the Eating Habits Scale, and the Portuguese Yale Food Addiction Scale 2.0. Thirty (22.2%) participants presented food addiction problems. The logistic regression models utilized suggest that participants in the group with food addiction problems are more likely to seek clinical help to control weight, to consider that they should eat less food high in sugar, and to report lower food adequacy. In sum, this finding highlighted a connection between food addiction, weight dissatisfaction, eating habits, and food choices in college students, a population at risk for developing and retaining eating pathologies. Further research is essential to evaluate and implement interventions regarding food addiction, weight dissatisfaction, eating habits, and food choices in college students.

## 1. Introduction

The construct of food addiction has received increasing research attention over the past decade. Despite the lack of consensus about the construct, a widely used definition of food addiction emerged by applying the criteria for substance dependence to eating behaviors. It postulates that certain foods, particularly highly processed, hyper-palatable, and caloric foods, may have an addictive potential similar to that observed in substance use disorders [1]. This consumption stimulates the dopaminergic brain circuits, leading to increased food ingestion and triggering symptoms related to abstinence [2]. In fact, one definition of addiction is: a compulsive pattern of seeking and using addictive substances, even when possessing the awareness of their potential for harm. Furthermore, consumption may be accompanied by several negative emotional states following abstinence and a high propensity to relapse [3]. Such symptoms are common to substance and food addiction and seem to be related to brain networks for reward, stress, and executive control, as has been observed and modulated in studies registering brain activation and neuromodulation, respectively [4,5].

The concept of food addiction usually appears to be connected with other eating behaviors that are characterized by an excessive/abnormal food intake [6]. However, there is still disagreement about the state of food addiction as a phenomenon separated from eating disorders. Some evidence supports that food addiction shares symptomatology with binge eating disorders, but other studies suggest that they are two distinct problematics [7,8]. Regardless, the prevalence of food addiction is considerably higher in individuals with an eating disorder diagnosis and/or in individuals with overweight/obesity when compared to their non-clinical counterparts [6,9,10]. Thus, these data support the role of food addiction in association with eating and weight psychopathology.

In non-clinical populations, food addiction appears to be more prevalent in college students (24%) than in the general population (20%) [6,11]. College students are a recognized risk group for eating pathologies. Previous studies suggest that the transition to college frequently implies changes in eating habits. The individuals tend to move from patterns rooted within the family to more autonomous food choices and, therefore, new eating habits. Specifically, these times are typically marked by lack of food variety, skipping meals, intake of high caloric and high-fat food between meals, and food choices driven mostly by the taste of food and the pleasure of eating [12,13,14,15].

Overall, the excessive consumption of food leads to a reduction in the reward sensitivity of dopaminergic pathways, consequently leading to a loop of increased food intake, as it is seen in individuals with obesity, who often experience food cravings [16,17]. Accordingly, this unhealthy shift experienced during college years often leads to an increase of body weight [15], a variable that is associated with the presence of food addiction [11,18,19]. Considering that higher weight is associated with higher levels of body dissatisfaction [20], which, in turn, is related to maladaptive eating behaviors [21], it seems that the interconnection between these variables could contribute to a vicious circle.

Thus, it is possible to hypothesize that both eating behavior and weight-related variables contribute to justifying the high prevalence of food addiction among college students. However, to the best of our knowledge, no previous studies have specifically explored the relationship between food addiction, weight-related variables, eating habits, and food choices in college students. Hence, the present study aims to address this gap. Specifically, we propose two main goals: (1) to characterize food addiction in a sample of college students and (2) to compare college students with and without food addiction problems concerning weight-related variables, eating habits, and food choices. We hypothesize that participants with food addiction problems will present higher BMI, less weight satisfaction, unhealthier eating habits and food choices. To achieve our goals, we recruited a sample of college students to fulfill a set of online self-report questionnaires assessing the variables of interest.

## 2. Materials and Methods

### 2.1. Participants and Procedure

This cross-sectional study included college students from four higher education institutions in the North of Portugal (i.e., University of Minho, University of Porto, University of Beira Interior, and Catholic Portuguese University). The students could have been attending any course at the universities involved. The inclusion criteria were being aged between 18 and 35 years old and understand Portuguese language written and spoken.

College students were invited for participation between December 2020 and January 2021 (a period of lockdown due to COVID-19 in Portugal) through an institutional e-mail with information about the study and a link to respond to an online questionnaire. Participants were provided a link to Google Forms after virtually signing the informed consent of the study and answered several online questionnaires that assessed sociodemographic and anthropometric data, food choices, eating habits, and food addiction. This study was approved by the ethical review committee of the University of Minho (CEICSH 108/2020). The participants attending a psychology course at the University of Minho were associated with the Accreditation System for Participation in Experiences implemented by the School of Psychology, no other form of compensation was given to participants from other Universities and courses.

### 2.2. Measures

#### 2.2.1. Sociodemographic and Anthropometric Questionnaire

The Sociodemographic and Anthropometric Questionnaire was created for the present study to assess age, sex, marital status, level of education completed, university, course, height, weight, weight satisfaction, and eating pattern characterization (healthy or unhealthy eating).

#### 2.2.2. Food Choices Characterization

The Food Choices Characterization was a set of five questions created for the purpose of the present study that assessed the participants’ perception of their own eating as healthy or unhealthy and their ideal consumption of specific food. Specifically, participants were instructed to take into consideration their regular food choices and reflect whether they should ideally eat more, eat less, or eat the same of each type of the following foods: fruit, vegetables, foods high in sugar, foods high in fat, and fast food.

#### 2.2.3. Eating Habits Scale (Escala de Hábitos Alimentares—Created and Validated for Portuguese Population [22])

The Eating Habits Scale consists of a 40-item self-report measure to assess eating habits through a 5-point Likert scale (1 = Never (zero times per week); 5 = Always (7 or more times per week)). It comprised 4 subscales: Food Quantity (7 items; e.g., “I snack between meals” (reversed item)), Food Quality (14 items; e.g., “I prefer cookies or cake to bread” (reverse item)), Food Variety (8 items; e.g., “I eat the same food all the time” (reverse item)), and Food Adequacy (11 items; “I usually eat breakfast”). The minimum possible score was 40 points, and the maximum possible score was 200 points. It is also possible to calculate the mean for both total scale and subscales (as we adopted in the present study), meaning that the scores range between 1 and 5. The higher the scores, the healthier the eating habits are considered. Additionally, the original authors considered 80 points as a cut-off to healthy eating habits (i.e., M_total scale_ > 2). This measure presents good internal consistency for the present study (Cronbach α = 0.79 for the total score).

#### 2.2.4. Portuguese Yale Food Addiction Scale 2.0 (P-YFAS 2.0; [18,23])

The Portuguese Yale Food Addiction Scale is a 35-item self-report measure that assesses food addiction in the last 12 months through an 8-point Likert scale (0 = Never; 7 = Everyday). The items were developed based on DSM-5 criteria for substance use disorders and adapted to assess 11 symptoms of food addiction. Namely, overeating, desire to cut down, time spent eating/searching for food, food craving, related impairment (such as avoiding or not performing well at work/school, family, or social relationships), risky use (physically hazardous, detrimental physical/psychological consequences), tolerance to food, and withdrawal. Each symptom was considered fulfilled when one or more of the relevant items reached the threshold (different cut-off for each parameter assessed, as defined in the validation of YFAS 2.0 [23]). Further, the food addiction diagnosis requires scoring at least one question related to the impairment or distress criteria. A final symptom count score is calculated to inform about food addiction severity by adding up all symptoms (Mild: 2–3; Moderate: 4–5 symptoms; and Severe: 6 symptoms). This measure presents good internal consistency in the present study (Cronbach α = 0.95).

### 2.3. Statistical Analysis

The statistical analyses were performed using IBM^®^ SPSS^®^ version 28 (Chicago, IL, USA). To perform the described analysis, we used the calculation of measures of central tendency and dispersion. Initially, an exploratory analysis was performed and data normality was tested. Subsequently, *t*-tests for independent samples were used to analyze the differences between the presence/absence of food addition problems and the other anthropometric and eating variables under study. The independence chi-square test (χ^2^) was used to analyze associations between dichotomic variables. All significant variables were included in the subsequent binary logistic regression analysis (hierarchical block method) to assess the predictors of being in the group with food addiction problems. To prevent overlap, two independent binary logistic regression models were used, one that considered the total score of the Eating Habits Scale and the other contemplating only its significant subscales. A *p* value of <0.05 was determined to indicate statistical significance.

## 3. Results

The participant sample was composed of 194 college students aged between 18 and 32 years old (M = 20.85, SD = 2.78). Most participants were from the University of Minho (75.7%) and enrolled on a bachelor’s degree course (62.4%). Additionally, most were single (96.6%) and females (89.2%). The mean self-reported BMI was 22.27 (SD = 3.64). A more detailed characterization of the sample can be found in Table 1.

Thirteen (6.7%) females of the total of participants presented diagnosis for food addiction (i.e., two or more food addiction criteria fulfilled plus the presence of the impairment/distress criteria on P-YFAS 2.0 [18,23]). Of these, two (15%) participants were classified as having mild food addiction, four (31%) having moderate food addiction, and seven (54%) having severe food addiction. Additionally, 17 other female participants (15.5%) confirmed at least two food addiction criteria but without the presence of clinically significant impairment or distress. These 30 participants (22.2%) were denominated “group with food addiction problems” and were analyzed together. Participants reporting less than two food addiction criteria were defined as “group without food addiction problems”. Figure 1 presents a flow diagram to represent the group division.

Table 2 displays the characterization of the two groups concerning the variables under study. Although there were no significant differences between the group with food addiction problems and the group without food addiction problems in relation to body mass index (BMI), there was a statistic significant association with weight satisfaction. Specifically, only 23.3% of the participants with food addiction problems were satisfied with their weight in comparison with 45.7% of the participants without food addiction problems. Accordingly, the participants of the group with food addiction problems were also more likely to have sought clinical help to control their weight.

Regarding the eating habits, the groups had significantly differed in the total score of the Eating Habits Scale. Specifically, the group with food addiction problems reported unhealthier eating habits when compared with the group without food addiction problems. Additionally, the group with food addiction problems reported significantly less Food Quality and Food Adequacy than the group without food addiction problems.

Accordingly, a perception of unhealthy eating patterns was reported by 60% of the participants in the group with food addiction problems against only 30.5% of the participants in the group without food addiction problems. Similarly, more participants of the group with food addiction problems reported that they should eat less food high in sugar (96.7%), food high in fat (66.7%), and fast food (63.3%), than participants of the group without food addiction problems (55.5%, 57.7%, and 68.3%, respectively).

Two binary logistic regressions with two blocks were performed to ascertain the likelihood of participants belonging to the group with food addiction problems (versus the group without food addiction problems) based on the variables that were statistically significant. The predictors/variables were divided into two groups: weight-related variables (i.e., weight dissatisfaction and previous clinical help to control weight) and eating-related variables (i.e., perception of eating pattern, food choices (ideal consumption) of food high in sugar, food choices (ideal consumption) of food high in fat, food choices (ideal consumption) of fast food, and eating habits). Concerning eating habits specifically, the Eating Habits Scale—Total Score was entered as predictor in the first model and the Subscales Food Quality and Food Adequacy were entered in the second model. For both models, the weight-related variables were entered first on block 1 and the eating-related variables were entered on the block 2. The results are summarized in Table 3.

In both logistic regression models, the weight-related variables were statistically significant, χ^2^(2) = 10.652, *p* < 0.001. According to the pseudo r-square, between 0.053 (Cox and Snell) and 0.093 (Nagelkerke) of the variability was explained by this set of variables. This model classified 84.5% of all cases. The previous clinical help to control weight was a significant predictor (OD = 5.258; 95% CI = [1.151; 6.026]), meaning that participants who sought clinical help to control their weight have a higher probability of belonging to the group with food addiction problems.

Concerning the model including the Eating Habits Scale, the logistic regression model continued to be statistically significant when the eating-related variables were added to the second block, χ^2^(5) = 29.551, *p* ≤ 0.001. The role of these variables produced a pseudo-r-square between 0.187 (Cox and Snell) and 0.324 (Nagelkerke) and accurately classified 85.1% of the cases. The ideal consumption of food high in sugar was a significant predictor for food addiction problems (OD = 6.469; 95% CI = [0.009, 0.541]), with the participants reporting that they should eat less food high in sugar presenting a higher probability of belonging to the group with food addiction problems. Then, the final logistic regression model suggests that participants in the group of food addiction problems are 2.90 times more likely to have sought clinical help to control weight and 0.07 times more likely to consider that they should eat less food high in sugar than the amount they usually eat.

In the model including the Subscales Food Quality and Food Adequacy from the Eating Habits Scale, when the eating-related variables were added to the second block, the logistic regression model also continued to be statistically significant, χ^2^(6) = 32.8231, *p* ≤ 0.001. The role of these variables produced a pseudo-r-square between 0.201 (Cox and Snell) and 0.348 (Nagelkerke) and accurately classified 86.6% of the cases. The ideal consumption of food high in sugar (OD = 7.538; 95% CI = [0.007, 0.433]) and the Food Adequacy Subscale (OD = 5.776; 95% CI = [0.052, 0.742]) achieve statistical significance as predictors. This means that participants that considered they should eat less food high in sugar and that reported lower food adequacy had a higher probability of belonging to the group with food addiction problems. Therefore, the final logistic regression model suggests that participants in the group of food addiction problems are three times more likely to have sought clinical help to control their weight, 0.05 times more likely to consider that they should eat less food high in sugar, and 0.20 times more likely to report lower food adequacy.

## 4. Discussion

This study sought to further understand food addiction in a sample of college students and its relationship with weight-related variables, eating habits, and food choices. Thirteen (6.7%) of the college students included in the study sample presented a diagnosis of food addiction and another 17 (15.5%), although they did not show the required clinically significant impairment or distress to present a diagnosis, fulfilled considerable food addiction criteria. Additionally, it was found a relationship between the presence of food addiction and weight dissatisfaction, unhealthier eating habits, and consumption of foods high in sugar/high in fat/fast-food.

Although previous systematic reviews placed the prevalence of food addiction between 0 and 25.7% for non-clinical populations [6,24,25], there is some controversy regarding the prevalence of food addiction depending on the population studied. The frequency of food addiction diagnoses found in this study sample (i.e., 6.7%) is in line with another study involving undergraduate college students in Spain which found a similar prevalence of food addiction (i.e., 6.4%; [26]). Additionally, more than half of the participants that fulfilled criteria for a diagnosis of food addiction in the present study were classified with severe diagnoses according to P-YFAS 2.0. Although similar results were consistently highlighted in the systematic review of Penzenstadler and colleagues (2019) [25] and also in the recent study with undergraduate students conducted by Romero-Blanco and colleagues (2021) [26], no explanations have yet been advanced. One possibility is that YFAS 2.0 could be a more sensitive measure to identify severe cases. Another speculation is that severe cases of food addiction may have been rising due to the current obesogenic environment of the modern world (an obesogenic environment is characterized by the high availability of food at any time of the day [27]).

Notwithstanding, fewer severe food addiction symptoms may potentially lead to adverse outcomes [18,23,28,29]. For this reason, in the group with food addiction problems we choose to include both participants with a full diagnosis of food addiction and those who fulfilled the food addiction criteria without the presence of clinically significant impairment or distress. Thus, the high frequency of 22.2% found in the presents study seems to corroborate the predisposition to eating pathology during college years [12,13,14,15].

Only females reported food addiction problems. This is likely due to the fact that our sample was composed mostly by female participants (89.2%). Considering that food addiction is more prevalent in females than in males [6,19,24,25], the low number of male participants in our sample may explain the absence of males with food addiction in our study. Additionally, some evidence suggests that gender differences in hormone profiles and dietary patterns put women at more risk than men to engage in problematic eating behaviors [30].

Contrary to most of the previous evidence showing that BMI is positively associated with food addiction [11,18,19], the present study did not find significant BMI differences between the group with food addiction problems and the group without food addiction problems. However, our sample was composed mostly of normal-weight participants (81.4%), which may have prevented this association. Nevertheless, we found evidence that linked food addiction problems and weight dissatisfaction. Similarly, the participants who had previously engage with clinical help to control their weight were more likely to present food addiction problems. Although having clinical health support to control weight is positive, it makes sense to assume that people with weight dissatisfaction would be more likely to seek clinical help to address it. Therefore, we believe that such evidence is not a causal correlation but a reflex to some degree (present or past) of weight dissatisfaction. While the role of body/weight dissatisfaction in the development and maintenance of general eating pathology is well-established [31], less is known so far about the specific relationship between body (dis)satisfaction and food addiction. A previous study by Şanlier and colleagues (2016) [32] investigated related variables and founded a negative relationship between body image and food addiction. Along these lines, other study reported that body shame increases as the severity of food addiction [33]. Contradictorily, Usta and Pehlivan (2021) [34] found no direct pathway between body dissatisfaction and food addiction. Thus, the present findings contribute to shedding some light on the controversial focus on body satisfaction as a buffer factor in the development of food addiction. Specifically, our results suggest that interventions targeting food addiction for this population must addressed body dissatisfaction and weight-related health issues.

Regarding eating habits, both groups reach the cut-off considered in the Eating Habits Scale [22], meaning that they are more prone to healthy eating habits. Two possible explanations can be proposed to address such findings: (1) The increased awareness about healthy nutrition along with the social responsibility trend in food behavior that young consumers have been demonstrating by paying more attention to the impact of their eating behavior on the environment [35,36]; and (2) the data collection occurred during a specific time frame in Portugal, when lockdowns due to COVID-19 were imposed and most of the classes were taking place online. Thus, although we did not assess information regarding COVID-19 issues in the present study, it is expected that the number of college students staying in their family home instead of having moved out and living alone/with colleagues was much higher during this time frame, which could have contributed to the healthier eating habits found. That being said, the available literature on the changes in eating habits during the COVID-19 pandemic suggests a general increase in snack frequency and a higher consumption of sweets and ultra-processed food [37]. So, although comprehending eating habits during COVID-19 was not the aim of our study, we must consider that the pandemic and its associated lockdowns could had influenced our findings both ways, by contributing to healthier or unhealthier eating habits.

Notwithstanding, the group with and without food addiction problems differ in their global eating habits, food quality, and food adequacy with the group with food addiction problems reporting unhealthier global eating habits, lower food quality and lower food adequacy than the group without food addiction problems. Specifically, food adequacy (i.e., characterized as the ability to meet standard guidelines/recommendations regarding eating habits) was a significant predictor of belonging to the group with food addiction problems. Accordingly, a perception of an unhealthier eating patterns was reported by most participants with food addiction problems. Similarly, most participants of the group with food addiction problems considered that they should reduce the consumption of food high in sugar, fat, and fast food. In particular, the intention to reduce the consumption of food high in sugar was a significant predictor of belonging to the group with food addiction problems. These findings are aligned with the predictions based on previous evidence stating that participants with food addiction problems have unhealthy eating habits, characterized by a substantial consumption of foods high in sugar, fat, and salt [1]. Simultaneously, it also provides further support to the claim that food addiction is a disordered eating behavior that deserves attention. Thus, it is needed to promote healthier eating in college students and to address food addiction in this population.

Some limitations should be considered, namely the preponderance of female participants that difficult generalization for both sexes. The discrepancies between the sample size of the two groups compared (i.e., 30 participants in the group with food addiction problems against 164 in the group without food addiction problems). Accordingly, the group with food addiction problems included not only participants with a diagnosis of food addiction but also participants that did not present criteria for a food addiction diagnosis (i.e., with clinically significant impairment/distress), preventing specific conclusions about the clinical sub-sample with diagnoses food addiction. The study included college students from several courses, and due to the sample size and variation of the number of students in each course, it was not possible to investigate differences according to university background. The study occurred during pandemic lockdowns due to COVID-19, which could have impacted the eating habits and food choices of the participants involved. However, with a cross-sectional design it is not possible to draw causal inferences. The use of self-report measures to evaluate complex constructs did not account for social desirability. Future studies should address the same variables with a larger sample size of individuals with food addiction problems/food addiction diagnoses, be composed of a more matched number of males and females, take into consideration the university course background, and follow a prospective design.

## 5. Conclusions and Policy Implications

Although college students are a population at risk for developing and maintaining eating pathology and weight gain, to the best of our knowledge, no previous study explored the relationship between food addiction and eating/weight variables for this population. The present study objectives were to characterize food addiction in a sample of college students and to understand the relationship between food addiction problems, weight-related variables, eating habits, and food choices. Thus, a sample of college students were recruited between December 2020 and January 2021 and fulfilled a set of self-report questionnaires assessing the variables of interest. Thirty (22.2%) of the one hundred and ninety-four college students included in the study presented food addiction problems. Additionally, previous clinical help to control weight, consumption of foods high in sugar, and lack of food adequacy predict the likelihood of having food addiction problems. However, some limitations should be noted: the preponderance of females, the low number of participants with food addiction problems, the application of only self-reported measures, and the use of a cross-sectional design that prevents causal inferences. Future studies should continue to explore these variables facing the highlighted limitations, aiming a further understanding of their interconnections and the development of potential interventions targeting food addiction, the relationship with self-weight, eating habits, and food choices in college students.

## Figures and Tables

**Figure 1 ijerph-19-14588-f001:**
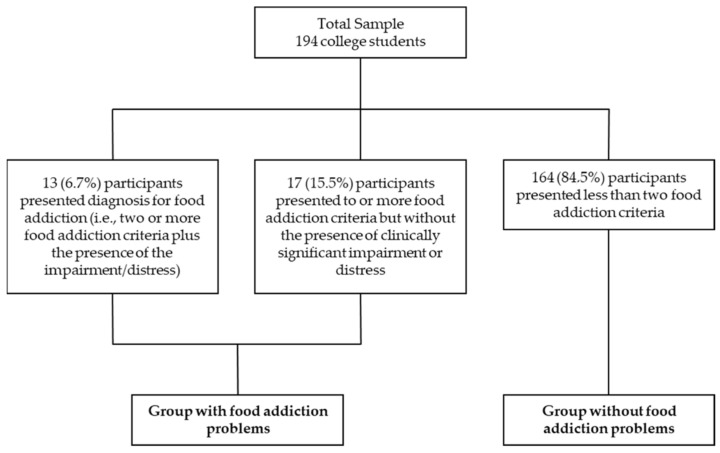
Flow diagram representing the division between the “Group with food addiction problems” and the “Group without food addiction problems” in accordance with P-YFAS 2.0.

**Table 1 ijerph-19-14588-t001:** Socio-demographic and anthropometric characterization of the study sample.

	X ± SD	*n* (%)
Age	20.85 ± 2.78	-
Sex		
Female	-	173 (89.2)
Male	-	21 (10.8)
Higher education institution		
University of Minho	-	146 (75.7)
University of Porto	-	13 (6.7)
University of Beira Interior	-	8 (4.1)
Catholic Portuguese University	-	27 (13.9)
Course	-	-
Psychology	-	165 (85.1)
Education	-	11 (5.7)
Medicine	-	5 (2.6)
Others	-	13 (6.7)
Marital status		
Single	-	188 (96.9)
Married/Registered partnership	-	6 (3.4)
BMI	22.27 ± 3.64	-
Weight status		
Underweight	-	9 (4.6)
Normal weight	-	158 (81.4)
Overweight/obese	-	4 (2.1)

Note. X = mean; SD = standard deviation; *n* = number of participants; % = relative frequency.

**Table 2 ijerph-19-14588-t002:** Characterization of the group with food addiction problems and the group without food addiction problems regarding the variables under study.

	Group with Food Addiction Problems (*n* = 30)	Group without Food Addiction Problems (*n* = 164)	*t*	*df*
	X ± SD	X ± SD		
BMI	22.49 ± 2.84	22.22 ± 3.78	0.146	192
Eating Habits Scale (Total)	3.33 ± 0.27	3.49 ± 0.30	2.916 **	192
Food Quantity	2.81 ± 0.54	2.89 ± 0.53	0.741	192
Food Quality	3.11 ± 0.31	3.30 ± 0.37	2.569 *	192
Food Variety	3.89 ± 0.46	4.03 ± 0.46	1.519	192
Food Adequacy	3.51 ± 0.35	3.75 ± 0.39	3.181 **	192
	***n* (%)**	***n* (%)**	**χ^2^**	** *df* **
Weight satisfaction				
Satisfied	7 (23.3)	75 (45.7)	5.214 *	1
Not satisfied	23 (76.7)	89 (54.3)		
Previous clinical help to control weight				
Yes	14 (46.7)	36 (22.0)	8.098 **	1
No	16 (53.3)	128 (78.0)		
Perception of eating pattern				
Healthy	12 (40.0)	114 (69.5)	9.703 **	1
Unhealthy	18 (60.0)	50 (30.5)		
Food Choices (ideal consumption)				
Fruit				
Eat less/the same	9 (30.0)	73 (44.5)	2.189	1
Eat more	21 (70.0)	91 (55.5)		
Vegetables				
Eat less/the same	9 (30.0)	65 (39.6)	0.998	1
Eat more	21 (70.0)	99 (60.4)		
Food high in sugar				
Eat less	29 (96.7)	91 (55.5)	18.226 ***	1
Eat the same/more	1 (3.3)	73 (44.5)		
Food high in fat				
Eat less	20 (66.7)	69 (42.1)	6.178 *	1
Eat the same/more	10 (33.3)	95 (57.7)		
Fast Food				
Eat less	19 (63.3)	51 (31.1)	11.427 ***	1
Eat the same/more	11 (36.7)	113 (68.3)		

Note. X = mean; SD = standard deviation; *n* = number of participants; % = relative frequency; *t* = *t*-test; χ^2^ = chi-square test; *df* = degrees of freedom; BMI = body mass index. * = *p* < 0.05; ** = *p* < 0.01; *** = *p* ≤ 0.001.

**Table 3 ijerph-19-14588-t003:** Binary logistic regression to ascertain the likelihood that participants belong to the group with food addiction problems attending to weight satisfaction, previous clinical help to control weight, eating habits, eating pattern, and food choices (ideal consumption) of food high in sugar, high in fat, and fast-food.

	Eating Habits Scale—Total Score						Eating Habits Subscales					
	Block 1			Block 2			Block 1			Block 2		
	ß	SE	Exp (B)	ß	SE	Exp (B)	ß	SE	Exp (B)	ß	SE	Exp (B)
Weight satisfaction	0.825	0.472	2.281	0.578	0.519	1.782	0.825	0.472	2.281 ^+^	0.476	0.527	1.609
Clinical help to control weight	0.968	0.422	2.634 *	1.065	0.470	2.901 *	0.968	0.422	2.634 *	1.083	0.479	2.953 *
Perception of eating patterns				0.109	0.551	1.115				0.115	0.546	1.122
Food high in sugar				−2.674	1.051	0.069 *				−2.926	1.066	0.054 **
Food high in fat				−0.476	0.537	0.621				−0.472	0.539	0.624
Fast food				−0.515	0.518	0.597				−0.504	0.519	0.604
Eating Habits Scale				−1.558	0.908	0.211 ^+^						
Food Quality Subscale										0.069	0.695	1.072
Food Adequacy Subscale										−1.626	0.676	0.197 *
Chi-square	10.652 **			29.551 ***			10.652 **			32.823 ***		
	40.203 ***						43.475 ***					

Note. ß = regression coefficient; SE = variation of the unstandardized regression weight; Exp (B) = odds ratio; dependent variables: group without food addiction problems (0) vs. group with food addiction problems (1); independent variables: weight satisfaction (0 = satisfied; 1 = not satisfied); previous clinical help to control weight (0 = no; 1 = yes); eating patterns (0 = healthy; 1 = unhealthy); food high in sugar (0 = eating less; 1 = eating the same/more); food high in fat (0 = eating less; 1 = eating the same/more); fast food (0 = eating less; 1 = eating the same/more). ^+^ = *p* < 0.10; * = *p* < 0.05; ** = *p* < 0.01; *** = *p* ≤ 0.001.

## Data Availability

The datasets generated and/or analyzed in the current study are available from the corresponding author on reasonable request.

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
