# Peer review of "Food Addiction Problems in College Students: The Relationship between Weight-Related Variables, Eating Habits, and Food Choices"

_ijerph, 2022, doi:10.3390/ijerph192114588_

Round 1

Reviewer 1 Report

Thank you for giving me the opportunity to review this manuscript entitled: "Food addiction problems in college students: The relationship with weight-related variables, eating habits and food choices", to be considered for publication at IJERPH. 

Below you will find my comments and suggestions, which I believe can improve the quality of the final document. I really find this manuscript interesting.

-Are students from a single campus?

- Abstract: Nothing to say here. The information is accurate. Perhaps the selected questionnaires could be included. 

- Introduction: Correct line 41 howsoever. Correct line 43, they mix two different referencing styles. 

- Eating Habits Scale [13] put the name of the instrument and who it is. Some information is missing here. 

- I am curious to know what the students studied, have you considered that there might be a difference between the degrees? Perhaps this is a future line of research. 

- Results, line 181, instead of putting p = .0005 value the same format as in the table <.001.

- I think both the introduction and the discussion could be a little more elaborate. It seems to me that the results obtained are compared with few previous studies, and the number of references in the manuscript could be higher. 

- I have detected some minor errors in the bibliography.

Otherwise, nothing to highlight.

Reviewer 2 Report

The topic of the paper is quite interesting, but the paper is written so poorly. I want to give chance to authors to revise it significantly in line with my suggestions given below:

1. In the introduction, the authors should highlight the importance of this study, the objectives of the study, describe the methods used, and final conclusions.  

2. In the introduction, the authors should add all major points of innovations of this study. They should discuss the research gap, and how this study fills that gap.

3. The authors have totally missed writing the literature about the topic. I strongly suggest adding it.

4. In this paper, authors have analyzed the data with so simple statistical tools, which are not enough for publishing in peer-reviewed journals. I strongly suggest applying more rigorous economic techniques such as Analytical Hierarchy Process (AHP) or others for analyzing the data.

5. The last section should be "Conclusion and Policy Implications". The sequence should be like this 1) introductory sentences, purpose, model, sample period, data, a brief description of overall results, policy implications, and lastly limitations of the study and recommendations for future research.

Reviewer 3 Report

Dear Sirs,

Thank you very much for the opportunity to review the manuscript Food addiction problems in college students: the relationship with weight-related variables, eating habits and food choices.

The study aimed to better understand food addiction in a sample of college students and its relationship with weight-related variables, eating habits and food choices.

Undoubtedly, the topic is very important and deserves a detailed evaluation. All the more so since it concerns students, that is, a group particularly vulnerable to the risks of eating pathologies. The authors have made a good case for the need to conduct such a study.

Nevertheless, the manuscript requires minor additions and corrections.

First of all, the study was conducted during a lockdown period due to COVID-19, i.e. when students were in the family home. I don't know how long this period lasted in Portugal, but it's difficult to compare this period with the period of normal classes and students being physically in class. It is not clear from the methodology from what period the responses obtained from the students came from, while the introduction indicates that the survey is about students during the period they were away from home. This needs to be deciphered.

I provide detailed comments below:

Introduction

The study was conducted during the period when students were confined to their homes due to COVID-19 so the impact of this on students' eating behavior, habits and mental health should be described.

Materials and Methods

It would be helpful to present the results obtained from the Eating Habits Scale.

Given the specific conditions during this period (lockdown due to COVID-19), please specify what period the scale covered. Was it stated in the questionnaires that the survey covers the period when the students were away from the family home?

2.1 Participants and Procedure

From which years of study and which majors did the students come from?

How many of them lived away from home and how many of them lived in the family home/ Or whether they all stayed in the family home at the time.

It should be mentioned that the survey was conducted during the lockdown period due to COVID-19.

Have you calculated the minimum sample size? The size of the study group is small.

2.2.2 Food Choices Characterization.

How many questions did the questionnaire consist of? What was the possible scale of responses?

What does the ideal consumption of certain products mean? What scale did you adopt here and what guided you.

What period did the questionnaire take into account?

Line 99 What is the threshold value

Results

Please add a table with characteristics of study participants.

Line 127 for P-YFAS 2.0 please add the literature

Line 125-133 this is significant for this study so it would be helpful to present this graphically

139-141 please specify if this was statistically significant.

Table 1

Usually the mean and standard deviation are presented as X±SD and the median as Me(QR). Here is the mean? Presenting it in this form is misleading.

What is meant by t and x2- add in the legend below the table

Line 146-151- at what level was p.

Line 163-164-what does ideal consumption mean? Please describe here or in the methodology what thresholds you used to assess ideal consumption of high sugar, high fat and fast food?

Table 2

Please describe here ß SE Exp(B)

Line 180 with weight or Weight satisfaction?

184-186 This somewhat suggests that prior medical attention is threatening in the development of FA, please rephrase this a bit.

199- please briefly characterize these subscales in the methodology.

Discussion

Line 220 - What does severe diagnoses mean?

224 - On what basis did you assess that the environment of these students was obesogenic? And which environment are you referring to? The student's home or the family home? Please clarify this.

It would be good to have a broader discussion with reference to the literature.

Conclusions

289-291 This conclusion should refer to this study rather than being so general

References

Adjust the references to the requirements of the journal (abbreviations of journal names 5,17,20 or full names)

Round 2

Reviewer 2 Report

I have reviewed it again, The authors have significantly revised it, and I feel satisfied with it. So it can be accepted.